# Rapid cell-free forward engineering of novel genetic ring oscillators

**Henrike Niederholtmeyer[1†], Zachary Z Sun[2†], Yutaka Hori[3], Enoch Yeung[3], Amanda Verpoorte[1], Richard M Murray[2,3], Sebastian J Maerkl[1]\***

[1]Institute of Bioengineering, School of Engineering, École Polytechnique Fédérale de Lausanne, Lausanne, Switzerland; [2]Division of Biology and Biological Engineering, California Institute of Technology, Pasadena, United States; [3]Division of Engineering and Applied Science, California Institute of Technology, Pasadena, United States

**Abstract** While complex dynamic biological networks control gene expression in all living organisms, the forward engineering of comparable synthetic networks remains challenging. The current paradigm of characterizing synthetic networks in cells results in lengthy design-build-test cycles, minimal data collection, and poor quantitative characterization. Cell-free systems are appealing alternative environments, but it remains questionable whether biological networks behave similarly in cell-free systems and in cells. We characterized in a cell-free system the 'repressilator', a three-node synthetic oscillator. We then engineered novel three, four, and five-gene ring architectures, from characterization of circuit components to rapid analysis of complete networks. When implemented in cells, our novel 3-node networks produced population-wide oscillations and 95% of 5-node oscillator cells oscillated for up to 72 hr. Oscillation periods in cells matched the cell-free system results for all networks tested. An alternate forward engineering paradigm using cell-free systems can thus accurately capture cellular behavior.

**\*For correspondence:** sebastian. maerkl@epfl.ch

[†]These authors contributed equally to this work

## Introduction

A central tenet of engineering involves characterizing and verifying complex systems in a simplified environment (*Lu et al., 2009*). Electronic circuits are tested on a breadboard to verify circuit design and aircraft prototypes are tested in a wind tunnel to characterize their aerodynamics. A simplified environment does not exist for characterizing and engineering complex biological networks, requiring system analysis to be conducted primarily in cells. Performing extensive, quantitative and rapid network characterization in cells is limited due to difficulties associated with measuring parts, components, and systems in complex and ill-defined cellular hosts (*Kwok, 2010*). Particular problems include: I) lack of precise control over network component concentrations, II) unpredictable interactions and integration with host cell processes, III) cumbersome molecular cloning, and IV) technical challenges and limited throughput associated with single cell measurements.

Cell-free systems promise to be efficient and effective tools to rapidly and precisely characterize native and engineered biological systems to understand their operating regimes. Reconstituted biochemical systems have allowed the study of complex dynamic and self-organizing behaviors outside of cells such as switches, oscillators and pattern-forming regulatory networks (*Schwille and Diez, 2009*; *Genot et al., 2013*; *van Roekel et al., 2015*). Networks assembled from simplified biochemistries such as oligonucleotide polymerization and degradation reactions can produce complex behaviors such as oscillations and provide insights into the working principles of biological regulatory systems (*Genot et al., 2013*; *van Roekel et al., 2015*). While a high degree of abstraction and simplification makes it easier to analyze the underlying principles of biological networks, it becomes

**eLife digest** Engineers often use simplified models to test their ideas. For example, engineers test small-scale models of new airplane designs in wind tunnels to see how easily air flows by them. This saves the engineers the time and expense of building a full-sized aircraft only to learn it has serious design flaws.

The interactions of genes and proteins within living cells can be incredibly complex, and working out how a particular network works can take months or years in living cells. To try to speed up and simplify the process, scientists are developing models that do not involve cells. These models replicate the chemistry inside of the cells and allow scientists to observe complex interactions between genes, proteins and other cellular components. Some scientists have recreated complex patterns of gene expression in these cell-free models, but these systems still take a long time to make. It is also not yet clear whether these models accurately depict what happens in living cells.

Now, Niederholtmeyer, Sun et al. have created a cell-free system that allows the interactions of a large network of genes to be examined in a single day – a process that would previously have taken weeks or months. To test the model, Niederholtmeyer, Sun et al. recreated how networks of genes in the bacterium *Escherichia coli* interact to form "oscillations", which produce a regular rhythm of gene expression. When the cell-free oscillator networks were inserted into live *E. coli* cells, the oscillators continued to produce the same patterns of gene expression as they did outside the cells.

Overall, the experiments show that cell-free models can accurately reproduce, or emulate, the behavior of cellular networks. This work now opens the door for engineering ever more complex genetic networks in a cell-free system, which in turn will enable rapid prototyping and detailed characterization of complex biological reaction networks.

challenging to implement more complex networks and to directly transfer results and networks between the cell-free and the cellular environment. Implementation of genetic networks in transcription-translation reactions has gained considerable traction because they rely on the cellular biosynthesis machinery and are compatible with a broad range of regulatory mechanisms. A growing number of synthetic gene networks with increasing complexity have been implemented in cell-free transcription-translation systems (*Noireaux et al., 2003*; *Shin and Noireaux, 2012*; *Takahashi et al., 2015*; *Pardee, 2014*). We and others have recently shown that oscillating genetic networks can be implemented in vitro outside of cells using microfluidic devices (*Niederholtmeyer et al., 2013*; *Karzbrun et al., 2014*). However, whether these cell-free systems reflect the cellular environment sufficiently well to be of significance to biological systems engineering and analysis remains an open question. A few studies investigated whether individual components such as promoters and ribosomal binding sites express at comparable strengths in cell-free systems and in cells (*Sun et al., 2014*; *Chappell et al., 2013*). Comparisons of the behavior of genetic networks in cell-free systems and in cells are still limited to a few examples such as repressor-promoter pairs (*Chappell et al., 2013*; *Karig et al., 2012*) and a RNA transcriptional repressor cascade (*Takahashi et al., 2015*). Thus far, however, it has not been shown whether genetic networks with complex dynamic behavior function similarly in cell-free and cellular environments.

Here, we demonstrate that cell-free systems can be used to characterize and engineer complex dynamic behaviors of genetic networks by implementing and characterizing novel 3-node, 4-node, and 5-node negative feedback architectures in vitro. We go on to show that our 3- and 5-node oscillator networks were functional in cells and that their periods were comparable to those observed in the cell-free system, indicating that the cell-free system accurately emulated the cellular environment for the complex dynamic networks developed and tested in this study. Cells carrying our 3-node networks oscillated on a population level, which was previously only observed in actively synchronized oscillators using quorum-based coupling (*Danino et al., 2010*). The in vitro and in vivo oscillations of our 5-node negative feedback networks confirm theoretical predictions on biomolecular ring oscillators and represent the largest synthetic negative feedback networks implemented simultaneously in cell-free systems and in cells.

## Results

### Cell-free systems allow rapid prototyping of novel genetic networks

Cell-free expression systems prepared from an *E. coli* extract ('TX-TL') can preserve the endogenous *E. coli* transcription machinery as well as native mRNA and protein degradation mechanisms (*Shin and Noireaux, 2010a*; *2010b*; *2012*; *Sun et al., 2013*). We have recently described a microfluidic nano-reactor device capable of emulating cellular growth and division. A discontinuous flow of TX-TL reagents through miniaturized reactors at rates matching natural *E. coli* dilution rates keeps transcription and translation rates at constant steady state levels and removes reaction products, which is critical for the implementation of dynamic genetic networks (*Niederholtmeyer et al., 2013*). Using a TX-TL expression system in the microfluidic nano-reactor device thus provides a simplified and controlled environment, which can be applied to rapid prototyping (*Sun et al., 2014*) and characterization of genetic networks (*Figure 1*, top, *Figure 1—figure supplement 1*). In addition, TX-TL provides significant time savings over traditional prototyping in cells (*Figure 1*, bottom, *Figure 1—source data 1*). Linear DNA can be used in *lieu* of plasmid DNA, and the DNA source does not require specific compatible origins of replication or antibiotic markers. Therefore, DNA can be assembled completely in

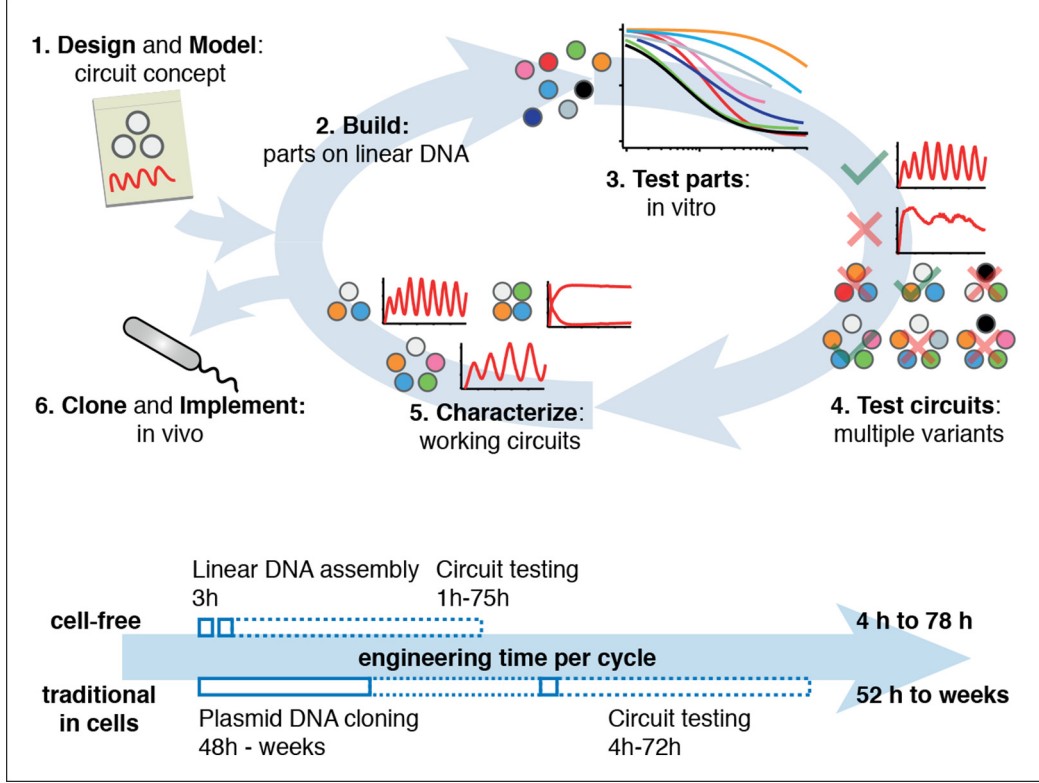

**Figure 1.** Cell-free systems allow rapid and extensive characterization of biological systems. Schematic representation of the design-build-test cycle using the cell-free system (top). A design is first modeled to obtain intuition about the architecture. Parts are then assembled on linear DNA without cloning, and tested in vitro. With functional parts, circuit variants can then be tested and working circuits can be extensively characterized. Final circuits are cloned onto plasmids and implemented in vivo. For a specific example of the cell-free system applied to engineering a 5-node oscillator network see *Figure 1*. Bottom shows a comparison of the time required for testing a genetic circuit by the cell-free approach versus traditional engineering in cells.

The following source data and figure supplements are available for figure 1:

**Source data 1.** Comparison of a Test Cycle in TX-TL vs. a Test Cycle in traditional prototyping.

**Figure supplement 1.** Engineering a 5-node negative feedback oscillator using the cell-free framework.

vitro based on premade modules within 3 hr (*Sun et al., 2014*). Importantly, the time needed does not scale with circuit complexity, as each circuit component can be added as a separate linear template, which can be easily replaced or varied in concentration in following design-build-test cycles. This is particularly advantageous for large circuits, where dosage or part changes require extensive re-cloning of plasmids for cellular testing, but no additional time for in vitro testing. We estimate that prototyping a genetic circuit in TX-TL requires 4 to 78 hr, as opposed to prototyping in *E. coli,* which requires a minimum of 52 hr to several weeks.

## Cell-free characterization of a synthetic in vivo circuit

We asked whether this cell-free system could be used to run and characterize an existing synthetic in vivo circuit and chose to test the repressilator (*Elowitz and Leibler, 2000*) as a model circuit. We successfully implemented the original repressilator network in our cell-free system and observed long-term sustained oscillations with periods matching the in vivo study (*Figure 2, Video 1*). We compared the original repressilator to a modified version containing a point mutation in one of the

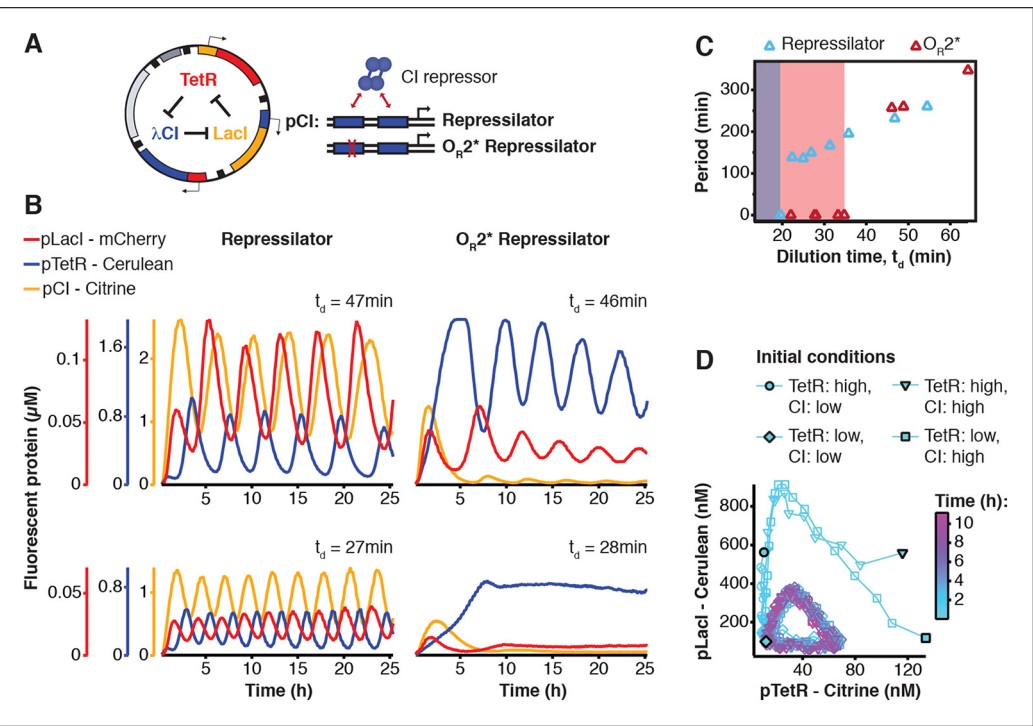

**Figure 2.** Cell-free repressilator characterization. (**A**) Application of cell-free systems to characterize the original repressilator (Elowitz and Leibler, 2000) and a modified version with a point mutation in the CI promoter (O$_R$2*) located in one of the binding sites of the CI repressor. (**B**) Expression from the three promoters of the repressilator and the O$_R$2* version at different dilution times. (**C**) Oscillation periods of the repressilator as a function of dilution time. In the O$_R$2* version sustained oscillations were supported in a narrower range of dilution times as compared to the original repressilator network. (**D**) Phase portrait of repressilator oscillations starting from different initial TetR and CI repressor concentrations.

The following figure supplements are available for Figure 2:

**Figure supplement 1.** Oscillation parameter regime for a 3-node repressilator network in terms of dilution time and synthesis rates.

**Figure supplement 2.** Existence of oscillations and periods for a 3-node repressilator network in terms of dilution time.

**Figure supplement 3.** Oscillation parameter regime for a 3-node repressilator network in terms of dilution time and repressor binding affinity

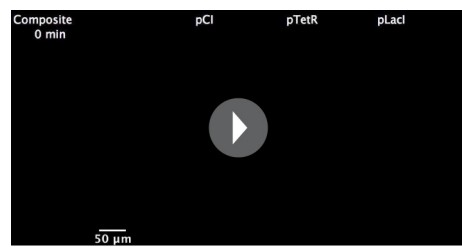

**Video 1.** 3-color in vitro run of repressilator at $t_d$ = 47 min. Shown on the right are individual channels of pCI-Citrine-ssrA, pTetR-Cerulean-ssrA, and pLacI-mCherry-ssrA. These are combined in the composite at the left.

CI repressor binding sites in the promoter regulating LacI (**Figure 2A**). This mutation increases the repressor concentration necessary for half-maximal repression ($K$), and reduces cooperativity (**Rosenfeld et al., 2005**). At long dilution times ($t_d$) both circuits oscillated, but with shifted absolute reporter protein concentrations (**Figure 2B**). At decreasing dilution times amplitudes decreased and periods became faster with a linear dependence on $t_d$. Faster dilution times, however, did not support oscillations for the modified network (**Figure 2B,C**). Experimentally, the range of dilution times supporting oscillations can serve as a measure for robust oscillator function, which generally diminishes with decreasing synthesis rates or when binding of one repressor to its promoter is weakened as in the $O_R2^*$ mutant (**Figure 2—figure supplements 1–3**). Initial conditions can influence the dynamic behavior of nonlinear systems but are difficult to control in cells. In order to explore the dynamics of the repressilator in response to different initial conditions we varied the starting concentrations of TetR and CI repressor. For all conditions tested the system quickly approached limit cycle oscillations and was invariant to initial conditions (**Figure 2D**). This analysis of the repressilator network in phase space provides an example for an experimental characterization that would be challenging or impossible to perform in a cellular environment.

## Engineering novel negative feedback circuits

The cell-free system also allows rapid characterization of individual network components. We measured the transfer functions of repressor-promoter pairs in the repressilator network (**Figure 3A**, **Figure 3—figure supplements 1–3**, **Figure 3—source data 1**) and found that the network is symmetric in terms of transfer functions. In the CI promoter $O_R2^*$ mutant we observed the expected shift in $K$ value and decreased steepness of the transfer function. We also characterized TetR repressor homologs as building blocks for novel negative feedback circuits (**Figure 3A**) and with the exception of QacR observed similar transfer functions as observed in vivo (**Stanton et al., 2014**) (**Figure 3—figure supplement 3**).

Using three new repressors, BetI, PhlF and SrpR, we constructed a novel 3-node (3n) circuit, 3n1, and observed high-amplitude oscillations over a broad range of dilution times with the same dependence of amplitude and period on $t_d$ as for the repressilator (**Figure 3B**). In our characterization of the repressilator network and the 3n1 oscillator we found dilution rates to be critical for the existence, period and amplitude of oscillations. Protein degradation is similar to dilution in that it results in removal of repressor proteins. In order to study the effect of degradation we constructed a second 3n network (3n2) using TetR, PhlF and SrpR repressors on linear DNA. One version of the circuit used strong ssrA ClpXP degradation tags, while the second used untagged repressors. We observed oscillations for both circuits (**Figure 3C**). However, the circuit without ssrA-tag mediated protein degradation exhibited slower oscillations, which extended to lower dilution times, showing that protein degradation, just like dilution, affects oscillator function and period. Effects of ClpXP-mediated protein degradation, which have been shown to be important for existence and frequency of oscillations in vivo (**Cookson et al., 2011**; **Prindle et al., 2014**), can thus be emulated in a cell-free environment. We characterized the repressilator (**Figure 2**) and the novel 3n1 network (**Figure 3B**) on plasmid DNA, reasoning that this is the closest approximation to the situation in a cell and because promoter strengths compare better when measured on a plasmid (**Sun et al., 2014**; **Chappell et al., 2013**). However, for a more rapid analysis of novel networks and network variants it is advantageous if laborious cloning steps are not required to obtain initial results on circuit performance. Construction and comparison of two 3n2 network variants, which only required a few PCR reactions to synthesize the linear DNA templates, showed that it is possible to go from theoretical design of a circuit to first experimental results in a very short timeframe (**Figure 3C**). Next, we went on to test this concept for novel and more complex network architectures.

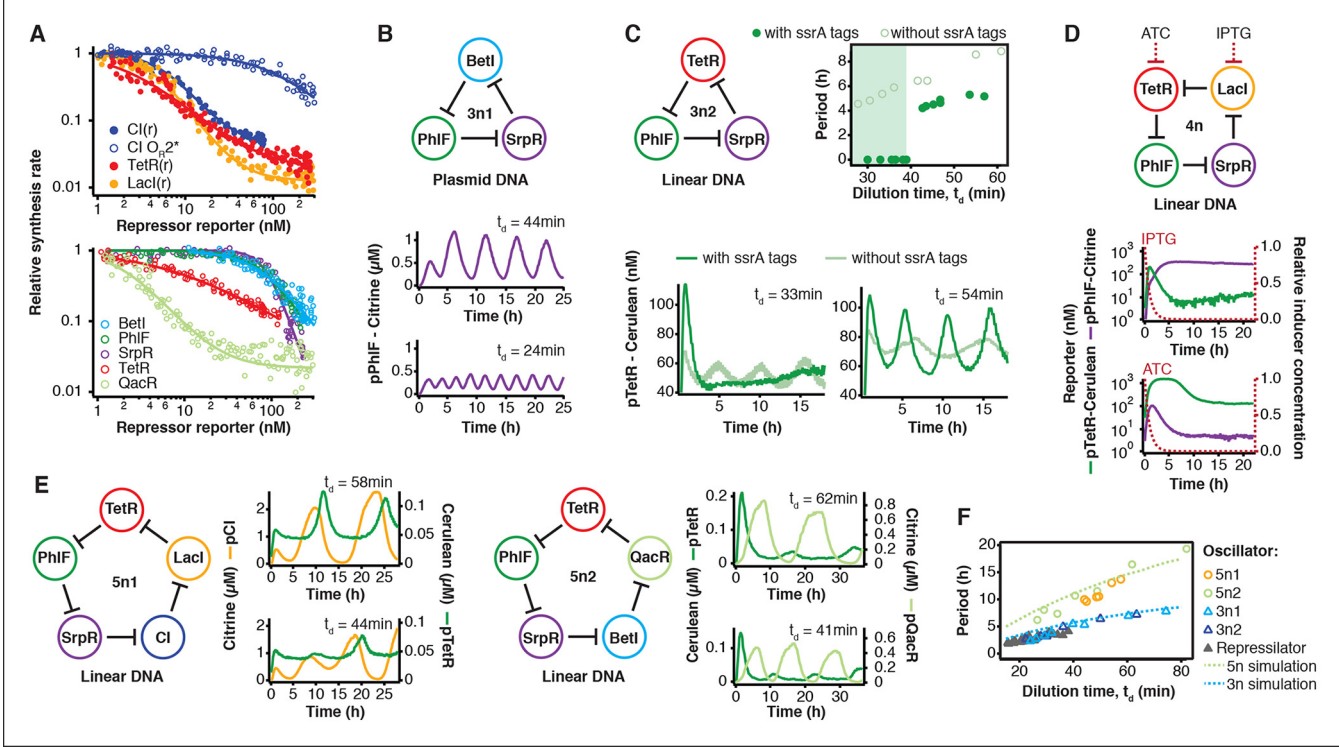

**Figure 3.** Cell-free prototyping and characterization of novel negative feedback circuits. (**A**) Transfer functions of the repressilator repressor-promoter pairs (top) and TetR homologs (bottom). The TetR repressor was tested against two different promoters: the promoter used in the repressilator (top panel) and the J23119-TetR promoter (*Stanton et al., 2014*) (bottom panel). Lines are Hill function fits. (**B**) Oscillations of a novel 3-node ring oscillator (3n1) constructed on plasmid DNA. (**C**) Two versions of a second 3-node ring oscillator (3n2) on linear DNA were used to study the effect of ClpXP degradation on oscillator function. One version was ssrA-tagged on all repressor genes while the other version did not carry degradation tags on the repressors. The same reporter with a medium-strength degradation tag was used in both versions. (**D**) A 4-node cyclic negative feedback network on linear DNA has two stable steady states that depend on the initial conditions. IPTG switched the network into the state where pPhlF was on and pTetR off. An initial pulse of aTc resulted in the opposite stable steady state. (**E**) Oscillations of two novel 5-node ring oscillators (5n1, 5n2) constructed on linear DNA. (**F**) 5-node ring oscillators oscillate with longer periods than 3-node ring oscillators, as predicted by simulations (Materials and methods) and shown by experimental data.

The following source data and figure supplements are available for figure 3:

**Source data 1.** Transfer function parameters. Parameter values of repressor – promoter pairs were determined by fitting to the Hill equation (Materials and methods). Promoter sequences were taken from the references cited.

**Figure supplement 1.** Measurement of transfer functions.

**Figure supplement 2.** Comparison of relative promoter strengths in vitro and in vivo.

**Figure supplement 3.** Comparison of half-maximal repressor concentrations needed for repression in vitro and in vivo.

---

Theory predicts that ring architectures built from an odd number of repressors oscillate, while even-numbered architectures have stable steady states (*Smith, 1987*; *Hori et al., 2013*). We experimentally built and tested a 4-node circuit from LacI, TetR, PhlF and SrpR on linear DNA. Initial pulses of LacI inducer IPTG or TetR inducer aTc allowed us to switch expression into either one of the two stable steady states (*Figure 3D*). Stable steady states were reached after an initial adjustment phase of 5 to 10 hr and remained stable until the experiment was terminated after 22 hr. These results show that non-oscillating networks also function in the cell-free environment and that the oscillations we observe for the 3-node networks are determined by network architecture and not established by particular reaction conditions in the microfluidic reactor.

Encouraged by the robust oscillations observed in the 3n networks and the expected behavior of the 4-node bistable switch, we built two 5-node ring networks (5n) to test our prototyping environment on another novel synthetic network architecture (*Figure 3E*). We expected these circuits to oscillate, as they were built from an odd number of repressors. Despite their considerable complexity both circuits indeed oscillated over a broad range of dilution times. The period of the 5n networks (up to 19 hr) was significantly longer than that of the 3n networks (up to 8 hr). Comparing all ssrA-tagged 3n and 5n ring architectures, we show that the observed periods could be accurately predicted for all four networks by computational simulations (*Figure 3F*). Our cell-free system allows characterization of complex networks from rapid testing on linear DNA to verifying networks cloned onto a single plasmid, which is the closest approximation to cellular implementation (*Figure 1—figure supplement 1*).

## Transfer of cell-free prototyped 3- and 5-node oscillators to *E. coli*

To validate our cell-free approach we cloned the 3n1 and 3n2 networks onto low-copy plasmids and co-transformed each with a medium-copy reporter plasmid into *lacI*-JS006 *E. coli* (*Stricker et al., 2008*). When tested on a microfluidic device (mother machine (*Wang et al., 2010*)), both 3n oscillators showed regular oscillations with periods of 6 ± 1 hr for at least 30 hr (*Figure 4A*, *Videos 2,3*). Both oscillators were surprisingly robust as all cells undergoing healthy cellular division oscillated (n=71) (*Figure 4—figure supplement 1*, *Video 4*).

We next turned to testing our 5n oscillators in vivo. Like the 3n networks we cloned both 5n oscillators onto low-copy number plasmids to transfer the two networks we had initially prototyped on linear DNA (*Figure 3E*) into *E. coli*. *Figure 1—figure supplement 1* shows the complete cell-free prototyping cycle for the 5n1 oscillator from design by testing on linear DNA to validation of the final cloned oscillator plasmid. In *E. coli,* 5n1 was not viable when co-transformed with a high expression-strength reporter, but was viable with a low expression-strength reporter. Specifically, 5n1 with a high expression-strength reporter caused slow growth and high cell death rates when run on the mother machine – we hypothesize that this is due to loading effects from high protein production, as decreasing the reporter expression strength resolved cell viability issues (*Ceroni et al., 2015*). When tested with a low expression-strength reporter both 5n oscillators showed robust oscillations in *E. coli* that were maintained for at least 70 hr, and over 95% of all analyzed traps containing healthy cells oscillated (n=104). In addition, both 5n networks oscillated with similar periods: 8 hr for 5n1, and 9 hr for 5n2 (*Figure 4B*, *Figure 4—figure supplement 1*, *Videos 5–7*).

We also tested both 3n oscillators on a CellASIC system, which allows planar single-layer colony formation. Starting from a single cell we observed striking oscillation pulses of the entire growing microcolony (*Figure 4C*, *Figure 4—figure supplement 2*, *Videos 8,9*). These population level pulses were also apparent when using three different fluorescent reporters simultaneously (*Figure 4—figure supplement 3*, *Video 10*). We did not observe population level oscillations in either the original repressilator, the $O_R2*$ mutant (*Figure 4—figure supplement 4*) or the 5n networks. Synchronized oscillations were not reported with the original repressilator (*Elowitz and Leibler, 2000*), and have only been observed in oscillators using intercellular communication (*Danino et al., 2010*; *Prindle et al., 2012*). In contrast to these quorum-sensing mechanisms that can actively couple and synchronize oscillator states (*Danino et al., 2010*; *Prindle et al., 2012*) we do not believe the population-wide in-phase oscillations we observed are due to an active coupling mechanism. We hypothesize that the population-level oscillations of the 3n1 and 3n2 networks are due to increased repressor concentrations as compared to the original repressilator network, which increases the inheritance of the period phenotype and minimizes the rapid de-phasing expected from stochastic cellular protein fluctuations (*Kiviet et al., 2014*). However, a quantitative characterization of this slow de-phasing phenotype requires more in depth understanding of stochastic effects in vivo.

Because cells stayed synchronized, we were able to analyze the population as a whole to make general conclusions of oscillator behavior. We varied dilution time by using different media conditions and media flow rates, and found a direct relationship between division times and period, consistent with the cell-free data collected. Oscillation periods of the 5n oscillators were also consistent with our cell-free results and showed a similar dependence on doubling time (*Figure 4D*).

Finally, we compared 3n1 and 5n2 with weak and strong reporters in vivo to analyze the effect of protein degradation on the oscillator period. We theorized that given a constant concentration of ClpXP, stronger reporters would result in more ClpXP loading, thereby slowing the period of

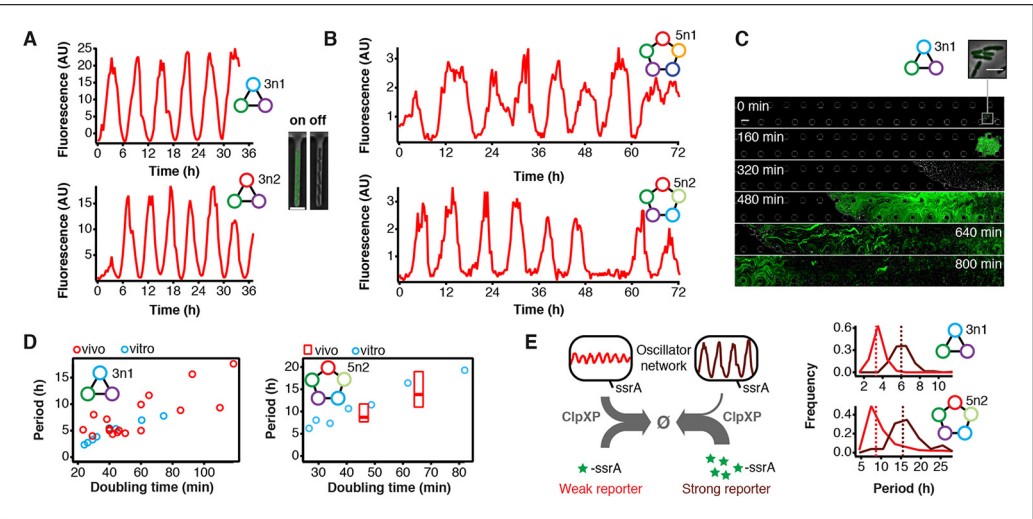

**Figure 4.** Novel 3-node and 5-node ring oscillators in cells. (**A**) Time series traces of 3-node ring oscillators running in *E. coli* (mother machine). Single trap traces of 3n1 and 3n2 observed for 36 hr in vivo using a strong pPhlF sfGFP-ssrA reporter and a representative image from an 'on' and 'off' state of oscillation. Scale bar: 5 µm. (**B**) Time series traces of 5-node ring oscillators running in *E. coli* (mother machine). Single trap traces of 5n1 and 5n2 observed for 72 hr in vivo using a weak pPhlF sfGFP-ssrA reporter. (**C**) 3n1 displays population-wide oscillation pulses in vivo (CellASIC). Time series micrographs of 3n1 under a strong pPhlF sfGFP-ssrA reporter every 160 min; inset shows individual cells of the initial microcolony. Scale bar: 10 µm and 5 µm (inset). (**D**) Relationship between period and division time in vivo. Left, 3n1 in vivo under a strong pPhlF sfGFP-ssrA reporter. The in vitro data is shown for comparison. Each point in the in vivo data corresponds to the period and division time from aCellASIC experiment run under different media type and flow rates. Right, 5n2 in vivo under aweak pPhlF sfGFP-ssrA reporter. In vivo periods determined at 29°C and 21°C growth temperature in mother machine experiments. Boxes represent the inner quartile range with the median. (**E**) Influence of reporter concentration on oscillation periods by competing for ClpXP degradation. Left, with constant amounts of ClpXP the reporter concentration affects repressor degradation and thus oscillation period. Histograms of the periods observed with a weak and a strong pPhlF sfGFP-ssrA reporter for both 3n1 and 5n2 run in the mother machine. Dashed lines indicate the medians.

The following figure supplements are available for Figure 4:

**Figure supplement 1.** Robust oscillations of 3-node and 5-node oscillators in vivo.

**Figure supplement 2.** Population-level oscillations of 3n2 oscillator in vivo.

**Figure supplement 3.** Three-color oscillations and population-level oscillations of 3n2 oscillator in vivo.

**Figure supplement 4.** Original repressilator and OR2* mutant repressilator do not show population-level oscillations.

---

oscillation. ClpXP is thought to influence oscillation dynamics in vivo in this manner (*Cookson et al., 2011*). We found that in the mother machine, both the period distributions of 3n1 and 5n2 showed this characteristic (*Figure 4E*), which reflects our cell-free findings of differential –ssrA tag dependent period length (*Figure 3C*).

## Discussion

We showed that synthetic dynamic networks can be readily implemented, characterized, and engineered in a cell-free system and subsequently transferred to cellular hosts (*Figure 1*). Our results demonstrate the utility of this approach for biological systems engineering and component characterization. The cell-free system resulted in the experimental validation of previous theoretical predictions on even- and odd-numbered cyclic negative feedback circuits (*Smith, 1987*; *Hori et al., 2013*) and enabled the in vivo implementation of robust 5-node genetic oscillators. The cell-free

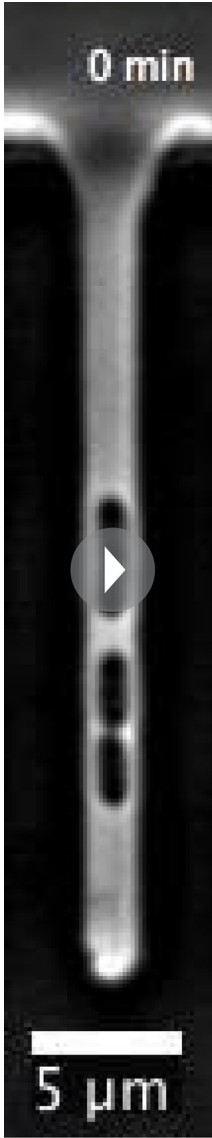

**Video 2.** 3n1 in mother machine, single trap. 3n1 using a pPhlF-BCD20-sfGFP-ssrA (strong) reporter is run in the mother machine at 29°C in LB.

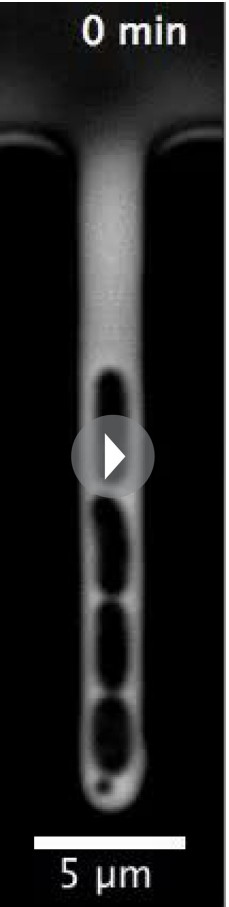

**Video 3.** 3n2 in mother machine, single trap. 3n2 using a pPhlF-BCD20-sfGFP-ssrA (strong) reporter is run in the mother machine at 29°C in LB.

environment can thus fill the gap between theoretical network design and in vivo implementation of biological systems and provide a simplified and controlled environment that drastically reduces the design-build-test cycle (*Sun et al., 2014*). In order to match the cell-free and cellular environments as closely as possible we prepared an extract-based TX-TL expression system from the same *E. coli* strain we used for in vivo experiments by a method that preserves the endogenous biosynthetic and protein degradation machinery (*Shin and Noireaux, 2010a*; *2010b*; *2012*; *Sun et al., 2013*). Similar expression systems had previously been applied to the prototyping of promoters and ribosomal binding sites (*Sun et al., 2014*; *Chappell et al., 2013*). Here we showed that they can also be used for the prototyping of entire genetic networks when they are analyzed in a microfluidic system that emulates cellular growth. Oscillation periods in negative feedback ring oscillators are mainly determined by degradation and dilution rates (*Hori et al., 2013*), which explains the good correspondence between oscillation periods in our cell-free environment and in *E. coli*. By constructing and testing novel network architectures we show that almost all prototyping can be done on linear DNA, which requires less than 8 hr to assemble and test. This allows rapid screening of different network topologies and rapid screening of parameters important for the desired function of a network. We observed some differences between the cell-free and cellular environment, particularly in the difficulty of predicting cellular toxicity and loading effects of the 5n oscillators in vivo, and

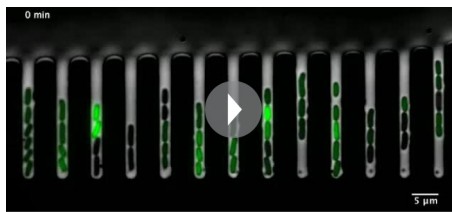

**Video 4.** Run of a panel of 3n2 oscillators in mother machine, using pPhlF-BCD20-sfGFP-ssrA (strong) reporter. DOI: 10.7554/eLife.09771.023

some differences between promoter and repressor strengths. With a better understanding of loading effects cell-free prototyping environments may predict when cells will be overloaded. We also add that the complete cell-free prototyping cycle that worked for 5n1 (*Figure 1—figure supplement 1*) was not entirely successful for the 5n2 network. 5n2 showed similar oscillations as compared to 5n1 when analyzed on linear DNA in the cell-free environment (*Figure 3E*) and on plasmid DNA in *E. coli* (*Figure 4*), but we did not observe oscillations for 5n2 when run from the same plasmid DNA in the cell-free environment. We hypothesize this is due to differences in expression efficiencies from linear and plasmid DNA in cell-free prototyping environments (*Sun et al., 2014*; *Chappell et al., 2013*), and/or differences between repressor strengths in the cell-free and the cellular environment (particularly for QacR, *Figure 3—figure supplement 3*). While more work is necessary describing and explaining differences between in vitro and in vivo environments, the observed behavior of complex networks in our cell-free environment reflected network behavior in vivo well. Cell-free systems are thus a powerful emulator of the cellular environment allowing precise control over experimental conditions and enabling studies that are difficult or time consuming to perform in

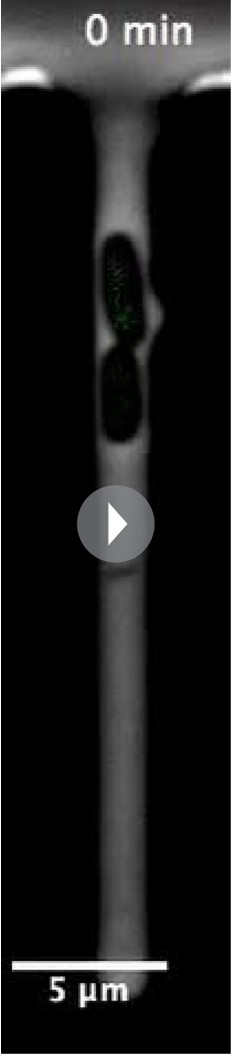

**Video 5.** 5n1 in mother machine. 5n1 using a pPhlF-BCD22-sfGFP-ssrA (weak) reporter is run in the mother machine at 29°C in LB. DOI: 10.7554/eLife.09771.024

cells. We envision that it will not only be useful for prototyping and characterizing novel synthetic systems but also facilitate in-depth analyses of native biological networks in a simplified setting. With further developments in cell-free lysate systems and supporting technologies, the cell-free approach is posed to play an increasing role in biological systems engineering and provides a unique opportunity to design, build, and analyze biological systems.

## Materials and methods

### DNA and strain construction

DNA was constructed using either Golden Gate Assembly or Isothermal Assembly. For linear DNA, all DNA was constructed using previously published Rapid Assembly protocols on a 'v1-1' vector (*Sun et al., 2014*). Linear DNA constructs are summarized in *Supplementary file 1A*. The original repressilator plasmid, pZS1 (*Elowitz and Leibler, 2000*) was used as a template for initial characterization and for construction of the O$_R$2* mutant. Transfer function plasmids were constructed by Transcriptic, Inc. For other plasmids, partial sequences were either obtained from Addgene

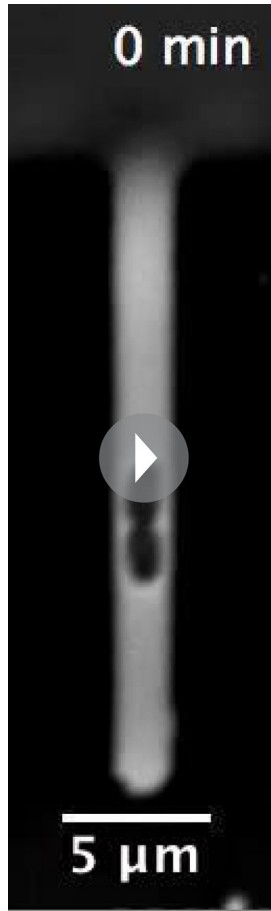

**Video 6.** 5n2 in mother machine. 5n2 using a pPhlF-BCD22-sfGFP-ssrA (weak) reporter is run in the mother machine at 29°C in LB.

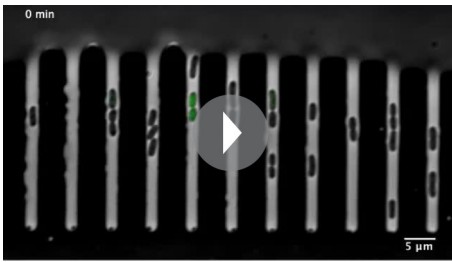

**Video 7.** Run of a panel of 5n2 oscillators in mother machine, using pPhlF-BCD22-sfGFP-ssrA (weak) reporter.

(*Stanton et al., 2014*) or synthesized on gBlocks or ssDNA annealed oligonucleotides (Integrated DNA Technologies). Specific plasmids required secondary-structure free segments, which were designed by R2oDNA (*Casini et al., 2014*). JS006 (*Stricker et al., 2008*) was co-transformed with origin-of-replication compatible plasmids to create engineered strains. Specifically, negative-feedback oscillator units were cloned onto pSC101* low copy plasmids (ampR or kanR), while reporters were cloned onto colE1 medium copy plasmids (kanR or cmR) (*Supplementary files 1B,C*). To modulate the reporter copy number, all experiments were conducted below 37°C (*Fitzwater et al., 1988*). Strain passage was minimized to avoid plasmid deletions due to the *recA* + nature of JS006 and the high complexity of oscillator plasmids or tri-ple-reporter plasmid. Based on the in vitro and in silico results, we used strong transcriptional and translational (*Mutalik et al., 2013*) units to maximize gain.

## TX-TL reactions

Preparation of TX-TL was conducted as described previously (*Sun et al., 2013*), but using strain 'JS006' co-transformed with Rosetta2 plasmid and performing a 1:2:1 extract:DNA:buffer ratio. This resulted in extract 'eZS4' with: 8.7 mg/mL protein, 10.5 mM Mg-glutamate, 100 mM K-glutamate, 0.25 mM DTT, 0.75 mM each amino acid except leucine, 0.63 mM leucine, 50 mM HEPES, 1.5 mM ATP and GTP, 0.9 mM CTP and UTP, 0.2 mg/mL tRNA, 0.26 mM CoA, 0.33 mM NAD, 0.75 mM cAMP, 0.068 mM folinic acid, 1 mM spermidine, 30 mM 3-PGA, 2% PEG-8000. For experiments uti-lizing linear DNA GamS was added to a final concentration of 3.5 μM (*Sun et al., 2014*).

## Steady-state reactions

Experiments were performed in a microfluidic nano-reactor device as described previously (*Niederholtmeyer et al., 2013*; *Sun et al., 2013*) with some modifications to optimize the condi-tions for the lysate-based TX-TL mix. Reaction temperature was 33°C. Lysate was diluted to 2x of the final concentration in 5 mM HEPES 5 mM NaCl buffer (pH 7.2). The reaction buffer mix was com-bined with template DNA and brought to a final concentration of 2x. For a 24 hr experiment 30 μl of these stocks were prepared. During the experiment, lysate and buffer/DNA solutions were kept in separate tubing feeding onto the chip, cooled to approximately 6°C, and combined on-chip. We ran experiments with dilution rates (μ) between approximately 2.8 and 0.5 hr$^{-1}$, which corresponds to dilution times, $t_d = \ln(2)\ \mu^{-1}$, between 15 and 85 min. These were achieved with dilution steps

**Video 8.** 3n1 in CellASIC. 3n1 using a pPhlF-BCD20-sfGFP-ssrA (strong) reporter is run in CellASIC. This video corresponds to *Figure 4C*. Conditions: 29°°C, LB media, 12.5% lamp intensity, 200 ms exposure, 2 psi flow rate.

**Video 9.** 3n2 in CellASIC. 3n2 using a pPhlF-BCD20-sfGFP-ssrA (strong) reporter is run in CellASIC. This video corresponds to *Figure 4—figure supplement 2*. Conditions: 29°C, LB media, 12.5% lamp intensity, 200 ms exposure, 2 psi flow rate.

exchanging between 7 and 25% of the reactor volume with time intervals of 7 to 10 min, which alternately added fresh lysate stock or fresh buffer/DNA solution into the reactors. Dilution rates were calibrated before each experiment. Initial conditions for the limit cycle analysis of the repressilator network were set by adding pre-synthesized repressor protein at the beginning of each experiment. For this, CI repressor (together with Citrine reporter) and TetR repressor (together with Cerulean reporter) were expressed for 2.5 hr in batch. On chip the initial reaction was mixed to be composed of 25% pre-synthesis reaction and 75% fresh TX-TL mix and repressilator template DNA. Then, the experiment was performed at a $t_d$ of 19.2 ± 0.3 min. Initial conditions for the 4-node experiment were 2.5 µM aTc or 250 µM IPTG, and the experiment was performed at a $t_d$ of 44.5 ± 0.9 min. DNA template concentrations used in steady-state reactions are listed in *Supplementary file 1D*. Arbitrary fluorescence values were converted to absolute concentrations from a calibration using purified Citrine, Cerulean, and mCherry, which were prepared using previously published protocols utilizing a His6 purification method followed by size-exclusion chromatography and a Bradford assay to determine protein concentration (*Sun et al., 2014*).

## Transfer function measurement

Transfer functions of the repressor – promoter pairs were determined in the nano-reactor device at a minimum of two different dilution times (*Figure 3—figure supplement 1*). All tested promoters were cloned into a plasmid in front of a BCD7 ribosomal binding site and the Citrine open reading frame. A non-saturating concentration of 1 nM plasmid was used in the experiment. The repressors were expressed from linear templates carrying the J23151 promoter and the BCD7 ribosomal binding site with time-varying concentrations, which were increased from 0 to 2.5 nM and decreased back to 0 during the course of the experiment (*Niederholtmeyer et al., 2013*). Simultaneously we expressed Cerulean as a reporter for the repressor concentration from a linear template at an identical concentration as the repressor template. From the concentration of the Citrine reporter we calculated the synthesis rate of the fluorescent protein over time using a model of steady state protein synthesis in the nano-reactor device (*Niederholtmeyer et al., 2013*),

$$P_d(t+\Delta t) = P_d(t) + syn(t) \cdot \Delta t - mat \cdot P_d(t) \cdot \Delta t - dil \cdot P_d(t) \tag{1}$$

$$P_f(t+\Delta t) = P_f(t) + mat \cdot P_d(t) \cdot \Delta t - dil \cdot P_f(t) \tag{2}$$

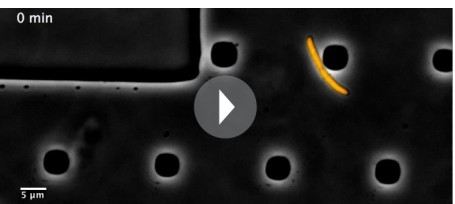

**Video 10.** 3n2 with 3-color output run in CellASIC. 3n2 using a pPhlF-BCD20-Citrine-ssrA, pTetR-BCD20-mCherry-ssrA, pSrpR-Cerulean-ssrA (strong) reporter is run in CellASIC. This video corresponds to Figure S5B. Conditions: 29 C, LB media, 12.5% lamp intensity, 200 ms exposure for Citrine and Cerulean (500 ms for mCherry), 2 psi flow rate.

where $P_d$ and $P_f$ are dark and fluorescent reporter concentration respectively, t is time, $\Delta$t is the time interval between dilution steps, *dil* is the volume fraction replaced per dilution step, which was determined during the calibration of the device, and *mat* is maturation rate of the fluorescent protein. Maturation times of Citrine and Cerulean were determined as described previously (*Niederholtmeyer et al., 2013*) and were 15 ± 4 min for Cerulean and 29 ± 3 min for Citrine. Dark fluorescent protein was calculated from (*Equation 2*):

$$P_d(t) = \frac{P_f(t + \Delta t) - P_f(t) + dil \cdot P_f(t)}{mat \cdot \Delta t} \tag{3}$$

and the synthesis rate was calculated from *Equation 1*:

$$syn(t) = P_d(t + \Delta t) - P_d(t) + mat \cdot P_d(t) \cdot \Delta t + dil \cdot P_d(t) \tag{4}$$

We used the sum of measured fluorescent Cerulean concentration and (*Equation 3*) for dark Cerulean as a measure of the total repressor protein present at any time during the experiment. The synthesis rates were normalized to their respective maximal values ($v_{max}$) and plotted against the concentration of the repressor reporter using only repressor concentrations higher than 1nM. The transfer curves were then fit to a Hill function

$$y = f(x) = y_{min} + (1 - y_{min}) \frac{K_M{}^n}{K_M{}^n + x^n} \tag{5}$$

where y is the synthesis rate, $y_{min}$ is the minimum synthesis rate, *n* is the Hill coefficient and $K_M$ is the Michaelis Menten constant for half maximal promoter activity. The fitting was performed in Igor Pro using orthogonal distance regression with ODRPACK95 assuming a 9% error in the measurements of Citrine and Cerulean fluorescence.

## $V_{max}$ measurements

Relative promoter strengths ($v_{max}$ values) were determined using the transfer function promoter plasmids. In vitro strengths were determined in 5 µl TX-TL reactions at a DNA template concentration of 1 nM. Reactions were assembled in 384-well plates, overlaid with 35 µl Chill-Out Liquid wax (BioRad) and analyzed using a Biotek SynergyMx plate reader set to 33°C reaction temperature, and reading Citrine fluorescence with Exc: $510 \pm 9$ nm and Em: $540 \pm 9$ nm. For comparison, Citrine fluorescence at 6 hr was normalized to the value of pLacI. In vivo strengths were determined using *E. coli* JS006 transformed with the same plasmids. Cells were grown at 29°C in MOPS medium supplemented with 0.4% glycerol and 0.2% casaminoacids. For each strain, three independent overnight cultures were diluted 1:50 and grown to mid-log phase. They were then diluted to a starting $OD_{600}$ of 0.15 into 100 µl growth medium in a 96-well plate and grown in the plate reader at 29°C with periodic shaking measuring Citrine fluorescence. Fluorescence values were normalized to OD resulting in steady state values after 2 hr. Average steady state values were normalized to pLacI for comparison with the in vitro measurement.

## In vivo experiments

Mother machine (*Wang et al., 2010*) experiments were conducted with custom-made microfluidic chips (mold courtesy of M. Delincé and J. McKinney, EPFL). *E. coli* cells were trapped in channels of 30 µm length, 2 µm width and 1.2 µm height. Before loading onto the device, cells were grown from a frozen stock to stationery phase. Cells were then concentrated tenfold and loaded onto the chip. Experiments were performed using LB medium supplemented with 0.075% Tween-20 at a flow rate of 400 µl/hr. Oscillation traces were collected from single mother machine traps using the background subtracted average fluorescence intensity of the entire trap.

CellASIC experiments were conducted using B04A plates (Merck Millipore, Darmstadt Germany). Flow rates were varied between 0.25 psi – 2 psi. Cells were grown from frozen stock in media at running temperature to stationery phase. Cells were then diluted 1:100 for 2 hr, and loaded on a equilibrated plate at 1:1000 or less to achieve single-cell loading efficiencies per chamber. To vary cellular doubling times, different growth media were used: LB (BD Biosciences), M9CA (Sigma Aldrich) with 0.2% glucose, 2xYT (MP Bio), MOPS EZ Rich (Teknova).

Cells were imaged in time series every 10-–20 min using a 100x phase objective minimizing both lamp intensity (12% Xcite 120, Excelitas Inc. Waltam MA or 1–2% CoolLED pE-2, Custom interconnected Ltd., UK) and exposure times (<500 ms) to limit photo-toxicity.

## Analysis of in vivo data

Images were processed and stitched (*Preibisch et al., 2009*), if necessary, using Fiji/ImageJ (*Schindelin et al., 2012*). Fluorescence traces of cell populations with synchronized oscillations were extracted from CellASIC movies using background corrected mean fluorescence intensity from the

entire field of view. For cells that were not synchronized over the complete field of view, we tracked regions of oscillating sister cells at the edge of the microcolony. We used ImageJ to define polygonal regions around those cells and manually shifted the polygonal region to track the front of growing cells. Periods were determined from fluorescence traces derived from mother machine and CellASIC movies by measuring the time from one oscillation peak to the next peak. Doubling times were estimated by averaging over the doubling times of at least ten individual cells.

## Model

We consider an $n$-node negative cyclic feedback biocircuit and denote the genes, mRNAs and proteins by $G_1, G_2, ..., G_n$, and $M_1, M_2, ..., M_n$ and $P_1, P_2, ..., P_n$, respectively. Let $r_i(t)$ and $p_i(t)$ denote the concentrations of mRNA $M_i$ and protein $P_i$, respectively. For example, the novel 3-node ring oscillator in **Figure 3B** is defined by n=3, $r_1(t) = $ [BetI mRNA], $r_2(t) = $ [PhlF mRNA], $r_3(t) = $ [SrpR mRNA], $p_1(t) = $ [BetI protein], $p_2(t) = $ [PhlF protein], $p_3(t) = $ [SrpR protein].

Our mathematical model considers transcription, translation and degradation of mRNA and protein molecules as summarized in **Table 1**, where $a_i$ and $b_i$ represent the degradation rates of $M_i$ and $P_i$, respectively, and $c_i$ and $\beta_i$ are the translation and transcription rates. The constants $K_{i-1}$ and $\nu_i$ are the Michaelis-Menten constant and the Hill coefficient associated with the protein $P_{i-1}$ and the corresponding promoter on gene $G_i$. We hereafter use subscripts 0 and $n + 1$ as the substitutes of $n$ and 1, respectively, to avoid notational clutter.

Using the law of mass action and the quasi-steady state approximation, the dynamics of the mRNA and protein concentrations can be modeled by the following ordinary differential equations (ODE)

$$r_\iota(t) = -(a_i + \mu)r_i(t) + \beta_i g \frac{K_{i-1}^{v_i}}{K_{i-1}^{v_i} + p_{i-1}^{v_i}(t)}, \tag{6}$$

$$p_\iota(t) = -(b_i + \mu)p_i(t) + c_i r_i(t),$$

where $i = 1, 2, ..., n$, and $g$ is the concentration of the circuit plasmid. The constant $\mu$ is the dilution rate of mRNA and proteins by the microfluidic device. The dilution time of the microfluidic device is defined by

$$T_d := \frac{\ln(2)}{\mu} \tag{7}$$

The ODE model (**Equation 6**) was numerically simulated using ode45 solver of MATLAB R2013b to obtain qualitative insight into the period as well as the oscillatory parameter regime (**Figure 3F** and **Figure 2—figure supplement 2**). The parameters summarized in **Table 2** were used for the simulations.

The plasmid concentration $g$ was set as $g = 5.0$ nM for **Figure 3F**. The initial concentrations for the simulations were $r_1(0) = 30$, $p_1(0) = 0$ and $r_i(0) = p_i(0) = 0$ for $i = 2, 3, ..., n$.

The period of oscillations was calculated based on the autocorrelation of the simulated protein concentration $p_1(t)$. More specifically, let

$$R(\tau) := \int_{T_1}^{T_2} p_1(t + \tau)p_1(t)\, \mathrm{d}t \tag{8}$$

**Table 1.** Stoichiometry and reaction rates.

| Description | Reaction | Reaction rate |
|---|---|---|
| Transcription of $M_i$ | $G_i + P_{i-1} \rightarrow G_i + P_{i-1} + M_i$ | $\beta_i \frac{K_{i-1}^{v_i}}{K_{i-1}^{v_i} + P_{i-1}^{v_i}}$ |
| Translation of $M_i$ | $M_i \rightarrow M_i + P_i$ | $c_i\, r_i$ |
| Degradation of $M_i$ | $M_i \rightarrow \emptyset$ | $a_i\, r_i$ |
| Degradation of $P_i$ | $P_i \rightarrow \emptyset$ | $b_i\, p_i$ |

**Table 2.** Parameters used for simulations.

| | Description | Parameter value |
|---|---|---|
| $a_i$ | Degradation rate of mRNAs (min$^{-1}$) | ln (2)/8 (half-life time: 8 min) |
| $b_i$ | Degradation rate of proteins (min$^{-1}$) | ln (2)/90 (half-life time: 90 min) |
| $\beta_i$ | Transcription rate (nM · min$^{-1}$ · plasmid concentration$^{-1}$) | 0.4 |
| $c_i$ | Translation rate (nM ·min$^{-1}$ · mRNA concentration$^{-1}$) | 0.5 |
| $K_i$ | Michaelis-Menten constant (nM) | 5.0 |
| $\nu_i$ | Hill-coefficient | 2.0 |

where $T_1$ is a positive constant such that $p_1(t)$ is steady state at $t = T_1$, and $T_2$ is a sufficiently large constant compared to the period of oscillations. The period of oscillations $T_{\text{period}}$ was determined by $T_{\text{period}} = \min_{\tau>0} \text{argmax}_{\tau} R(\tau)$. The simulation result is also consistent with the analytic estimation of the oscillation period in Hori et al. (*Hori et al., 2013*) in that the period increases monotonically with the dilution time $T_d$.

The parameter region for oscillations (*Figure 2—figure supplement 1*) was obtained based on the analysis result (Theorem 3) by Hori et al. (*Hori et al., 2011*). Since parameter values do not depend on the subscript $i$ as shown in the parameters table above, we remove the subscript $i$ and define $a := a_1 ( = a_2 = \dots a_n)$. In the same way, we define $b$, $c$, $\beta$, $K$ and $\nu$.

It was shown that the protein concentrations $p_i$ ($i = 1, 2, \dots, n$) oscillate if both of the following inequalities are satisfied (*Hori et al., 2011*).

$$\nu > W(n, Q), \tag{9}$$

$$c\beta > \left(\frac{W(n,Q)}{\nu - W(n,Q)}\right)^{\frac{1}{\nu}} \left(\frac{\nu}{\nu - W(n,Q)}\right) K(a+d)(b+d), \tag{10}$$

where

$$W(n,Q) := \frac{2\left(-\cos\left(\frac{\pi}{n}\right) + \sqrt{\cos^2\left(\frac{\pi}{n}\right) + Q^2\sin^2\left(\frac{\pi}{n}\right)}\right)}{Q^2\sin^2\left(\frac{\pi}{n}\right)} \quad and \quad Q := \frac{\sqrt{(a+d)(b+d)}}{(a+b+2d)/2}.$$

To obtain the parameter region in *Figure 2—figure supplement 1*, we substituted n =3 and the parameters shown in the table above into the right-hand side of the inequality condition (*Equation 10*), then we varied $T_d(= \ln(2)/\mu)$ between 5 to 80. The inequality (*Equation 9*) was always satisfied for these parameters.

The parameter region of *Figure 2—figure supplement 3* was obtained by the local stability analysis of the model (*Equation 6*). The previous theoretical result (*Hori et al., 2011*) showed that the model (*Equation 6*) has a unique equilibrium point and the protein concentrations $p_i$ ($i = 1, 2, \dots, n$) show stable oscillations if the Jacobian matrix evaluated at the equilibrium point has an eigenvalue in the open right-half complex plane. Based on this result, we computed the Jacobian eigenvalues with varying $K_3$, which we denote by $K_{cl}$, and $T_d$. The values in the parameters table above were used for the other parameters. The plasmid concentration was set as $g = 5.0$ nM in the computation.

## Acknowledgements

We thank Yin He, Transcriptic, Inc. and Holly Rees for cloning assistance, Jan Kostecki and Stephen Mayo for protein purification and size exclusion chromatography assistance, Rohit Sharma and Marcella Gomez for initial testing and modeling of oscillators in vitro, Kyle Martin for laboratory assistance, Adam Abate, Tanja Kortemme, and Charles Craik for laboratory space and equipment, Matthieu Delincé, Joachim De Jonghe, Marc Spaltenstein, John McKinney and Jin Park for mother machine material and assistance, Tim Chang and Benjamin Alderete for CellASIC assistance, and Michael Elowitz for insights and scientific support. This work was supported in part by EPFL and the

Defense Advanced Research Projects Agency (DARPA/MTO) Living Foundries program, contract number HR0011-12-C-0065 (DARPA/CMO). ZZS is also supported by a UCLA/Caltech Medical Scientist Training Program fellowship, ZZS and EY by a National Defense Science and Engineering Graduate fellowship, and YH by a JSPS Fellowship for Research Abroad. The views and conclusions contained in this document are those of the authors and should not be interpreted as representing officially policies, either expressly or implied, of the Defense Advanced Research Projects Agency or the U. S. Government.

## Additional information

### Competing interests

ZZS, RMM have ownership in a company that commercializes the cell-free technology utilized in this paper. The other authors declare that no competing interests exist.

### Funding

| Funder | Grant reference number | Author |
| --- | --- | --- |
| École Polytechnique Fédérale de Lausanne | | Henrike Niederholtmeyer<br>Amanda Verpoorte<br>Sebastian J Maerkl |
| Defense Advanced Research Projects Agency | MTO Living Foundries HR0011-12-C-0065 | Zachary Z Sun<br>Yutaka Hori<br>Enoch Yeung<br>Richard M Murray |

The funders had no role in study design, data collection and interpretation, or the decision to submit the work for publication.

### Author contributions

HN, ZZS, YH, EY, Conception and design, Acquisition of data, Analysis and interpretation of data, Drafting or revising the article; AV, Acquisition of data, Drafting or revising the article; RMM, Conception and design, Drafting or revising the article; SJM, Conception and design, Analysis and interpretation of data, Drafting or revising the article

## Additional files

### Supplementary files

• Supplementary file 1. Supplementary file 1A-D lists linear and plasmid DNA constructs, strains, and DNA concentrations used in cell-free experiments. (A) Linear DNAs used in this study. (B) Plasmids used in this study. (C) Strains used in this study. (D) DNA concentrations used in experiments.

• Source code 1. Zip folder containing the MATLAB code to produce all simulation results, specifically the periods of 3-node and 5-node networks at different dilution times and the oscillation parameter regimes in terms of synthesis rates, repressor binding affinity and dilution times (*Figure 3F*, *Figure 2—figure supplements 1-3*).

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
