## [Decision Letter]

Thank you for submitting your work entitled "Rapid forward engineering of novel genetic ring oscillators" for peer review at *eLife*. Your submission has been favorably evaluated by Naama Barkai (Senior editor), Friedrich Simmel (guest Reviewing editor), and two reviewers.

The reviewers have discussed the reviews with one another and the Reviewing editor has drafted this decision to help you prepare a revised submission.

Summary:

The article by Niederholtmeyer et al. demonstrates the realization of 3-node and 5-node genetic ring oscillators in a cell-free transcription/translation system, using a microfluidic nanoreactor with tunable dilution rate. After prototyping the oscillator circuits in vitro, they are implemented in *E.coli* and shown to function in vivo with similar dynamics, demonstrating a remarkable portability from in vitro to in vivo systems. This is important work, which significantly advances the state-of-the art in terms of circuit design and experimental approach. In particular, the article also contains the first realization of a synthetic 5-node genetic ring oscillator, which functions both in vitro and in vivo.

Essential revisions:

1) Biological significance:

The results themselves do not provide any new biological insight, but raise potentially interesting biological questions that could be addressed in more detail.

For instance, the ability of 3n1 and 3n2 constructs to show population level oscillations is biologically interesting, but unfortunately remains a mere observation with a speculative interpretation. It may also be misleading to term the observation "population level synchronization" (as in the last paragraph of the Main text). For synchronization, one would also expect evolution from an initially unsychronized state towards synchrony. In the present experiments, the bacteria start in a synchronized state and stay approximately synchronized due to slow dephasing (this interpretation is given in the subsection “Transfer of in vitro prototyped 3- and 5-node oscillators to *E. coli*”). Thus the observation might have a very different cause than the prototypical synchronization of coupled oscillators (as for quorum sensing-coupled bacteria).

In addition, the fact that the 5n networks had to be altered in order to function in vivo due to the system load is another avenue of continuing research that is left open ended by this current manuscript. Is it possible to add more biological insight?

2) Technical issues:

It is quite surprising that the approach taken by the authors actually works so well. Previous work on cell-free circuits did not show oscillatory dynamics (except for two publication cited in the text) because of low enzyme activity and long protein lifetimes, etc. It would be useful for the readers if the authors could point out the decisive technical steps that had to be taken for their work – the specific cell-extract used, the microfluidic dilution system, etc.?

In this context, can the authors comment on what they believe are the most important variables necessary to control for matching in vitro and in vivo environments?

Furthermore, the authors could more closely compare the time to implement designs both in their TX-TL systems and in cells? What is the true cost and time savings when working in vitro versus with *E. coli*?

3) Presentation:

The manuscript deals with variety of different experimental implementations of the oscillators, which can be slightly confusing at times. For instance, the 3n1 oscillator is implemented on a plasmid, while the 3n2 network is implemented on linear DNA. This is not motivated, however. Is this done for simplicity and faster prototyping?

Furthermore, it is not completely clear why the authors work with two different microfluidic systems for the experiments with bacteria – the "mother machine" (allowing simple single cell experiments?) and the CellASIC system (allowing single layer growth?). Figure 3 contains experiments with both of these two systems – can they be directly compared?

Figure 4 does a great job highlighting the main point of the paper, and this figure or a similar figure could come earlier in the manuscript to guide the reader towards the fact that speed is the true breakthrough. Showing/expanding Figure 4 to include what its in vivo only counterpart would look like, with associated time lines between the two methods, would be helpful to stress that speed is the main advantage here.

---

## [Author Response]

*Essential revisions: 1) Biological significance: The results themselves do not provide any new biological insight, but raise potentially interesting biological questions that could be addressed in more detail. For instance, the ability of 3n1 and 3n2 constructs to show population level oscillations is biologically interesting, but unfortunately remains a mere observation with a speculative interpretation. It may also be misleading to term the observation "population level synchronization" (as in the last paragraph of the Main text). For synchronization, one would also expect evolution from an initially unsychronized state towards synchrony. In the present experiments, the bacteria start in a synchronized state and stay approximately synchronized due to slow dephasing (this interpretation is given in the subsection “Transfer of in vitro prototyped 3- and 5-node oscillators to* E. coli*”). Thus the observation might have a very different cause than the prototypical synchronization of coupled oscillators (as for quorum sensing-coupled bacteria).*

We agree with the reviewers that the term “synchronization” was misleading. We did not want to imply that the population-wide oscillation we observed with 3n1 and 3n2 resulted from an active mechanism coupling the oscillator states in different cells. We removed the term “synchronization” throughout the manuscript and replaced it by “population-level” or “population-wide oscillations” to describe our observation. We extended the description of our observation and made the distinction from active synchronization mechanisms involving intercellular communication clearer. Just like the reviewer states, we explain the population-level oscillations we observed in growing microcolonies with a slow dephasing of cells that descended from a common mother cell. Our hypothesis is that this is due to increased repressor concentrations in cells carrying the 3n1 and 3n2 networks as compared to other oscillator networks for which we did not observe similar population-wide oscillation pulses. Higher protein concentrations would lead to reduced stochastic fluctuations during cell divisions and increase inheritance of the oscillator state from mother to daughter cells. We believe that these striking population-wide oscillations are an interesting observation that merits inclusion into our paper. With our focus on the engineering, prototyping and transfer of novel networks into cells it is however beyond the scope of our study to go into more detail on the stochastic effects that caused this interesting phenotype.

*In addition, the fact that the 5n networks had to be altered in order to function in vivo due to the system load is another avenue of continuing research that is left open ended by this current manuscript. Is it possible to add more biological insight?* Indeed, the 5n1 network required a lower-strength reporter in order to maintain cell viability in the mother machine. We hypothesize that this is the result of too much protein production, as decreasing the reporter strength resolved viability issues. Furthermore, we observed this trend with both 5n1 and 5n2, where cellular growth rates were slower than 3n1 and 3n2. We agree that this is an interesting point of further discussion, but defer to future work as loading effects in cells from synthetic circuits are still relatively poorly understood. However, with more understanding of loading effects one can theoretically implement those limitations into cell-free prototyping. We believe this is a rich area of future research with high value in the field. We have added to the manuscript the following to address this point:

To the Results:

“Specifically, 5n1 with a high expression-strength reporter caused slow growth rates and high cell death rates when run on the mother machine – we hypothesize that this is due to loading effects from high protein production, as decreasing the reporter expression strength resolved cell viability issues (2)”

To the Discussion:

“With a better understanding of loading effects cell-free prototyping environments may predict when cells will be overloaded.”

*2) Technical issues: It is quite surprising that the approach taken by the authors actually works so well. Previous work on cell-free circuits did not show oscillatory dynamics (except for two publication cited in the text) because of low enzyme activity and long protein lifetimes, etc. It would be useful for the readers if the authors could point out the decisive technical steps that had to be taken for their work – the specific cell-extract used, the microfluidic dilution system, etc.?*

We expanded the technical description of the TX-TL cell-free expression reagents and the microfluidic nano-reactor device we employ in the beginning of the Results section to make these points clearer. We believe that our approach worked so well because we combine a powerful expression system featuring high synthesis rates from *E. coli* promoters with precise dilution that the microfluidic device allows. Dilution allows synthesis to proceed at steady state solving the problems of declining enzyme activities and accumulation of proteins with long lifetimes. Preparation of the cell-extract is described in the Materials and methods in reference to previous work. In the revised version of our manuscript we added more details on the cell-extract into the main text and highlight more explicitly publications that described preparation, characterization and application of similar *E. coli* prototyping extracts. We also explain in more detail the function of the microfluidic device.

*In this context, can the authors comment on what they believe are the most important variables necessary to control for matching in vitro and in vivo environments?*

We would be happy to elaborate further. Previous work has described that it is possible to prototype promoters, ribosomal binding sites and small networks in cell-free environments. Preparation of the extract is important to preserve activity of the endogenous transcription and translation machinery and has been described previously. The extract also contains native enzyme responsible for mRNA and protein degradation. In combination with continuous dilution maintained by a microfluidic reactor device at rates that match dilution by cellular growth this results in removal of reaction products, which is critical for the implementation of dynamic genetic networks. It is also worth mentioning that we used the same *E. coli* strain to prepare the cell-extract as to perform the *in vivo* experiments. To supplement the expanded section on the decisive technical steps we took to match both environments in the beginning of the paper, we added to the discussion to give an explanation why periods of the oscillators matched so well in the cell-free and the cellular environment. We also added on the critical discussion to highlight that more work is necessary to describe and explain differences that exist between the in vitro and the in vivo environment we observed and had also already partly been described by others.

*Furthermore, the authors could more closely compare the time to implement designs both in their TX-TL systems and in cells? What is the true cost and time savings when working in vitro versus with* E. coli*?*

We agree that one of the strengths of the TX-TL system is the rapid turnover of experiments and data, and to address the point we’ve expanded the new Figure 1 to include a timeline for working in cell-free vs. with *E. coli*. In terms of cycle count, the table in Figure 1 includes “typical expectation.” We’ve added a section to the text to explicitly address the time savings when using TX-TL.

In terms of actual time to engineering in “PhD time”, we note that the collaboration between H. Niederholtmeyer and Z. Sun started in November 2013. The first proof of the repressilator functioning in vitro came in March 2014, and from March to July we characterized the repressilator. We successfully tested novel 3 node and 5 node architectures in vitro in a 1-month span, July 2014. The majority of the time, from August 2014 to the completion of the manuscript in April 2015, was spent transitioning our oscillator networks to in vivo compatible plasmids and diagnosing in vivo oscillations. Both H. Niederholtmeyer and Z. Sun estimate ½-person effort across the research period, thereby approximating 0.75-1 person-years to produce the results in the manuscript.

While we do not have direct experience building dynamic circuits solely in vivo, we believe that the 0.75-1 person-year estimate compares advantageously to other efforts that rely only on in vivo prototyping, as we found the majority of our time dedicated to diagnosing in vivo issues (cell viability, cloning issues, finding optimum imaging solutions, etc.).

*3) Presentation: The manuscript deals with variety of different experimental implementations of the oscillators, which can be slightly confusing at times. For instance, the 3n1 oscillator is implemented on a plasmid, while the 3n2 network is implemented on linear DNA. This is not motivated, however. Is this done for simplicity and faster prototyping?* In order to test if the cell-free environment could be potentially useful for prototyping of genetic networks with complex dynamics and transferring these between cell-free and cellular environments, we first tested the repressilator as a model circuit. Because this was an in vivo circuit it was located on a plasmid. When we found that it worked in the cell-free environment, we cloned the next novel network (3n1) onto a similar plasmid because this is the closest approximation to the situation in cells. When this novel network also worked, we decided to test even more novel network architectures. The real advantage of prototyping in a cell-free environment becomes clear when laborious cloning steps can be abolished for initial testing of networks, which is why we tested all further networks (3n2, 4n, 5n1, 5n2) first on linear DNA. Especially for large networks such as the 5-node oscillators the plasmid carrying the network becomes large and laborious to clone. In fact, if we had not found in our cell-free experiments with linear DNA that they had the potential to work in cells, we would not have attempted to implement them in *E. coli* and carried on when toxicity and loading effects further slowed cloning and in vivo testing.

Seen in this historical context of our experiments we believe that the different implementations of our networks make sense. We made this logic clearer in the manuscript highlighting the advantage of linear DNA for rapid prototyping and added the mode of implementation (“Plasmid DNA” or “Linear DNA”) to Figure 2 (now Figure 3) to avoid confusion. Additionally, we would like to point the reviewers to Figure 1–figure supplement 1 (former Figure 4—figure supplement 1), where we show on the example of 5n1 how we imagine that prototyping can be done first on linear DNA and then verified on plasmid DNA.

In addition to this logic of faster prototyping we have to add that 5n2 did not oscillate on plasmid DNA in the cell-free environment, whereas the network oscillated when implemented on linear DNA and on plasmid DNA in *E. coli*. We speculate that the reason why the 5n2 plasmid failed in vitro may be differences in expression efficiencies between linear and plasmid DNA that have been described previously, in combination with differences between repressor strengths in the cell-free and the cellular environment (particularly in QacR, see Figure 2—figure supplement 3). We included this example into our revised manuscript to show that while overall networks behaved very similarly in the cell-free and the cellular environment, more work is necessary to overcome some remaining technical challenges and to describe and explain differences that exist between the two environments.

*Furthermore, it is not completely clear why the authors work with two different microfluidic systems for the experiments with bacteria – the "mother machine" (allowing simple single cell experiments?) and the CellASIC system (allowing single layer growth?). Figure 3 contains experiments with both of these two systems – can they be directly compared?*

We found separate advantages and disadvantages to using either system. The reviewers are correct in that the mother machine allows for easier single cell experiments, while the CellASIC allows for observation of a population in a single layer growth environment. For experiments requiring single-cell analysis, we utilized the mother machine. CellASIC generally allows for more data collection, as we had the option of varying 4 variables in one run (vs. 1 variable for the mother machine). We therefore used these runs to populate the data in Figure 1, for 3n1, and to demonstrate population-level oscillations. However, we could not use the CellASIC to glean useful data for the 5-node oscillators, as oscillation periods were too long for the CellASIC runtime and cells did not demonstrate population-level oscillations. Therefore, all 5-node oscillator data was collected in the mother machine. We did not try to directly compare the two systems, but rather attempted to keep each panel consistent on its data collection method.

*Figure 4 does a great job highlighting the main point of the paper, and this figure or a similar figure could come earlier in the manuscript to guide the reader towards the fact that speed is the true breakthrough. Showing/expanding Figure 4 to include what its in vivo only counterpart would look like, with associated time lines between the two methods, would be helpful to stress that speed is the main advantage here.*

We thank the reviewers for the comments. We agree that Figure 4 would serve better as a summary slide, and that speed is important to emphasize. To that end, we have shifted Figure 4 to be Figure 1, added a bar to describe time, and added a source data table, which breaks down the time estimate. Please see responses to earlier comments for additional information.